# Associations between Plasma Folate and Vitamin B_12_, Blood Lead, and Bone Mineral Density among Adults and Elderly Who Received a Health Examination

**DOI:** 10.3390/nu14040911

**Published:** 2022-02-21

**Authors:** Ru-Lan Hsieh, Ya-Li Huang, Wei-Jen Chen, Hsi-Hsien Chen, Horng-Sheng Shiue, Ying-Chin Lin, Yu-Mei Hsueh

**Affiliations:** 1Department of Physical Medicine and Rehabilitation, Shin Kong Wu Ho-Su Memorial Hospital, Taipei 110, Taiwan; m001052@ms.skh.org.tw; 2Department of Physical Medicine and Rehabilitation, School of Medicine, College of Medicine, Taipei Medical University, Taipei 110, Taiwan; 3Department of Public Health, School of Medicine, College of Medicine, Taipei Medical University, Taipei 110, Taiwan; ylhuang@tmu.edu.tw; 4Department of Medicine, Section of Epidemiology and Population Sciences, Baylor College of Medicine, Houston, TX 77030, USA; wei-jen.chen@bcm.edu; 5Division of Nephrology, Department of Internal Medicine, School of Medicine, College of Medicine, Taipei Medical University, Taipei 110, Taiwan; 570713@yahoo.com.tw; 6Division of Nephrology, Department of Internal Medicine, Taipei Medical University Hospital, Taipei 110, Taiwan; 7Department of Chinese Medicine, Chang Gung University College of Medicine, Taoyuan 333, Taiwan; hongseng@ms1.hinet.net; 8Department of Family Medicine, School of Medicine, College of Medicine, Taipei Medical University, Taipei 110, Taiwan; green1990@tmu.edu.tw; 9Department of Geriatric Medicine, School of Medicine, College of Medicine, Taipei Medical University, Taipei 110, Taiwan; 10Department of Family Medicine, Wan Fang Hospital, Taipei Medical University, Taipei 110, Taiwan

**Keywords:** folate, vitamin B_12_, arsenic, lead, bone mineral density

## Abstract

This study hypothesized that plasma folate and vitamin B_12_ levels modified the association between blood lead and cadmium and total urinary arsenic levels and bone loss. A total of 447 study subjects who received a physical examination at the Wanfang Hospital Medical Center were recruited. Bone loss was defined as a calcaneus bone mineral density T-score less than −1. Blood cadmium and lead concentrations were measured by ICP-MS. Urinary arsenic species were determined using HPLC-HG-AAS. A SimulTRAC-SNB radioassay was used to measure plasma folate, vitamin B_12_, and homocysteine levels. Total urinary arsenic and blood lead concentration were positively correlated with the odds ratio (OR) for bone loss in a dose–response manner. The OR and 95% confidence interval (CI) for bone loss in participants with blood lead concentrations > 56.14 versus ≤33.82 μg/dL were 1.82 and 1.10–3.01. No correlation between plasma folate and vitamin B_12_ levels alone and bone loss was observed. However, this study is the first observational study to find that blood lead concentrations tend to increase the OR of bone loss in a low plasma folate and plasma vitamin B_12_ group with multivariate ORs (95% CI) of 2.44 (0.85–6.96).

## 1. Introduction

Osteoporosis, characterized as decreased bone mineral density (BMD), is associated with an increased risk of fractures and subsequent disability, and also contributes to increased morbidity and mortality in an aging population [1]. In a nationwide study analyzing data from the general population with ages equal to and over 50 years in the National Health Insurance Research Database of Taiwan, an increasing trend of osteoporosis prevalence has been reported in the decade from 2001 (17.4%) to 2011 (25.0%) [2]. The Taiwanese population is aging rapidly, and the prevalence of osteoporosis and related fractures has risen rapidly with age, posing an increasing threat to the elderly in Taiwan. Numerous factors have been identified as risk factors for osteoporosis, including sex, age, menopause, body mass index, alcohol consumption, tobacco smoking, corticosteroids use, and histories of fracture, diabetes, rheumatoid arthritis, and hyperthyroidism [3,4].

In addition to established risk factors, several environmental metal exposures, oxidants, and antioxidants are suspected to play a role in osteoporosis development. Previous evidence has implicated lead and cadmium, which may accumulate in the bone tissue of osteoporotic patients and be related to bone metabolism impairment [5]. Recently, a meta-analysis of fourteen published studies reported that exposure to lead and cadmium was associated with an increased risk of osteopenia or osteoporosis [6]. The positive association of cadmium exposure with osteopenia or osteoporosis was consistently observed in another meta-analysis that evaluated the association between urinary cadmium concentration and osteoporosis and osteopenia, but no association was found when assessing cadmium exposure in blood samples [7]. In a study of non-smoking postmenopausal women in Spain, no association was observed between bone health and exposure to lead and cadmium as assessed by the dietary intake questionnaire [8]. Therefore, the association between exposure to cadmium and bone mineral density remains inconclusive. Chronic arsenic exposure has also been identified as a risk factor for osteopenia [9]. Cell experiments have found that low-dose arsenic exposure was significantly associated with reducing the osteoblast differentiation of bone marrow cells [10]. In addition, alteration of bone microstructure and reduction of BMD were observed among rats that were exposed to 0.05 or 0.5 ppm arsenic in drinking water for 12 weeks [10]. However, few epidemiological studies have examined the association of arsenic exposure with low BMD.

Homocysteine, vitamin B_12_, and folate have been identified as determinants that may affect BMD. A meta-analysis has found that an increased level of homocysteine and vitamin B_12_ was observed among postmenopausal women with osteoporosis [11]. However, no association was observed between serum vitamin B_12_ and BMD in the study using data from the National Health and Nutrition Examination Survey (NHANES) in the United States [12]. Folate, a cofactor in homocysteine metabolism with vitamin B_12_ [13], has been found to be positively associated with higher BMD in the population of NHANES [12]. Nevertheless, in a Turkish study of postmenopausal women, serum levels of folate were not associated with BMD [14]. The association between vitamin B_12_, folate, and BMD remains inconclusive; more extensive studies are needed to clarify this association. The present study aimed to expand the evidence assessing the associations of environmental metal exposure, homocysteine, vitamin B_12_, and folate with bone loss, defined as decreased measured T-score of BMD. Given that environmental metal exposure has been implicated in homocysteine metabolism with respect to vitamin B_12_ and folate [15], we explored whether vitamin B_12_ and folate modify the association between environmental metal exposure and bone loss.

## 2. Materials and Methods

### 2.1. Study Subjects

A total of 405 subjects who received an adult health examination and 42 subjects who received a senile health examination at the Taipei Municipal Wanfang Hospital between July 2007 and September 2011 were enrolled in this cross-sectional study. Eligible participants included individuals who resided in Taipei City, spoke Mandarin, and expressed willingness to participate. In Taiwan, employees aged 40–65 can undergo a health examination each year provided by their designated hospital, which is a health and welfare measure provided by public or private organizations. All participants understood the purpose of the research and provided informed consent for urine and blood specimens to be collected for analysis in this study. The present study was approved by the Research Ethics Committee of Taipei Medical University, Taiwan (TMU-Joint Institutional Review Board N202007046) and conducted in accordance with the Declaration of Helsinki.

### 2.2. Questionnaire Interview and Specimens Collection

Each participant completed a standardized interview by a well-trained interviewer based on a structured questionnaire. The information collected included demographic and socioeconomic characteristics; lifestyle factors, such as cigarette smoking status, alcohol, tea, and coffee consumption; analgesic use; and personal medical history of hypertension and diabetes. Urine samples were collected at the time of recruitment and immediately transferred to a refrigerator at −20 °C for analysis of arsenic species. We used ethylenediaminetetraacetic acid (EDTA) vacuum syringes to collect 5–8 mL of peripheral blood samples and separated plasma and red blood cells at the time of recruitment and immediately transferred these to a refrigerator at −80 °C in order to prepare for the determination of homocysteine, folate, and vitamin B_12_, and red blood cells for the measurement of lead and cadmium concentrations.

### 2.3. Measurement of Bone Mineral Density

Quantitative ultrasound was used to obtain broadband ultrasound attenuation for measuring BMD in the calcaneus using an Ultrasound bone densitometer PEGASUS, Medilink. France. Quantitative ultrasound is a convenient and economical tool for evaluating BMD by quantifying the attenuations when ultrasound waves pass through the bone tissue at different speeds at the same time. A T-score was calculated by comparing the BMD of each participant to the peak BMD of a healthy 30-year-old adult. According to WHO standards, T-score ≤ −2.5 indicates osteoporosis, −2.5 < T-score < −1 indicates low BMD (osteopenia), and T-score ≥ −1 is considered normal BMD [16]. Since the number of participants with a T-score ≤ −2.5 was small in this study (*n* = 18), participants with a T-score ≤ −2.5 and −2.5 < T-score < −1 were collectively defined as the bone loss group in the analysis. Participants with a T-score ≥ −1 were defined as the normal bone mineral density group.

### 2.4. Measurement of Environmental Metal Exposure, Homocysteine, Vitamin B_12_, and Folate

We assessed inorganic arsenic exposure by summing the urinary concentrations of arsenite, arsenate, monomethylarsonic acid, and dimethylarsinic acid as quantified by a high-performance liquid chromatography-linked hydride generator and atomic absorption spectrometry [17]. Concentrations of total urinary arsenic were adjusted for urinary creatinine concentrations in order to take urinary dilution into account. Concentrations of lead and cadmium were measured from red blood cells determined by inductively coupled plasma mass spectrometry [18].

Concentrations of plasma homocysteine, vitamin B_12_ and folate were assessed with a radioassay kit (Bio-Rad, Richmond, CA, USA) and 1470 Wizard series gamma counters [19]. The validity, reliability, and detection limits of the measurement of metals, homocysteine, vitamin B_12_, and folate are presented in Appendix A.

### 2.5. Statistical Analysis

Means ± standard deviation and numbers (as percentages) were reported for continuous and categorical variables, respectively. The Wilcoxon rank-sum test and the Kruskal–Wallis test were employed to compare the differences between two groups or more than two groups for the continuous variables, respectively. The distributions of the categorical variables between groups were tested by means of the chi-squared test. The correlation between plasma folate and vitamin B_12_ levels and bone mineral density T-scores and plasma homocysteine levels were determined by a multivariate linear regression model after adjusting for confounding variables. Multivariate logistic regressions were used to analyze the association between the risk factors and bone loss. The cut-off points of the continuous variable among the independent variables were the corresponding tertiles of the reference group for these analyses. The risk of bone loss was calculated by multivariate-adjusted odds ratio (OR) and with a 95% confidence interval (CI). The significance test of the linear trend was to use the exposure stratification variable as a scoring variable and analyzed it as a continuous variable. All data analyses used the SAS software package (version 9.4; SAS Institute, Cary, NC, USA). Statistical significance was expressed as *p* < 0.05 (two-tailed).

## 3. Results

The association among sociodemographic characteristics, lifestyle, disease history, and bone loss are presented in Table 1. In total, 447 participants were aged 23 to 84 years. There were 284 males and 163 females. As age increased, the OR of bone loss increased, which indicates a significant dose–response relationship between the two factors. The OR of bone loss in women was higher than that in men. However, the difference was not significant. Furthermore, participants who frequently or occasionally drank alcohol had a significantly higher risk of bone loss than those who never drank, with an OR (95% CI) of 1.70 (1.10–2.63). Participants who frequently or occasionally drank coffee had a significantly lower risk of bone loss than those who did not drink coffee, with an OR (95% CI) of 0.64 (0.43–0.94). However, body mass index, education level, cigarette smoking habit, tea drinking, and history of diabetes or hypertension were not associated with bone loss.

As blood lead concentrations increased, the OR of bone loss significantly increased after adjustment for age, sex, and alcohol and coffee consumption. The OR (95% CI) of bone loss in patients with blood lead levels > 56.14 μg/dL was 1.82 (1.10–3.01) after multivariate adjustment compared to those with blood levels ≤ 33.82 μg/dL. Furthermore, a higher total urinary arsenic concentration was associated with a higher OR of bone loss (Table 2). No evidence of associations was observed between plasma folate, vitamin B_12_, homocysteine concentrations, and bone loss.

Notably, plasma homocysteine significantly decreased as plasma folate and vitamin B_12_ increased, with regression coefficients of −0.0197 (*p*-value = 0.0004) and −0.0012 (*p*-value = 0.014), respectively (Figure 1A,B). However, vitamin B_12_ and plasma folate and vitamin B_12_ were not related to BMD T-scores (Figure 1C,D).

These findings indicate that, although plasma folate and vitamin B_12_ concentrations are positively correlated (Figure 1E), they are not directly related to bone loss. To explore whether low plasma folate and low vitamin B_12_ levels alter the association between metals and bone loss, we performed a combination analysis. Patients with low plasma folate (≤6.95 ng/mL) and low vitamin B_12_ (≤522.50 pg/mL), low plasma folate and high vitamin B_12_, high plasma folate and low vitamin B_12_, and high plasma folate and high vitamin B_12_ concentrations were defined as low/low, low/high, high/low, and high/high groups, respectively. Concentrations of plasma homocysteine in the low/low and the low/high and high/low groups were significantly higher than those in the high/high group (Table 3). However, no difference was observed for total urinary arsenic or blood lead concentrations. In the low/low group, the total urinary arsenic concentrations for individuals with bone loss were significantly higher than those for individuals with normal bone mineral density. In the low/high and high/low and the high/high groups, the total urinary arsenic concentrations for individuals with bone loss were marginally higher than those for individuals with normal bone mineral density. Furthermore, in the low/low group, blood lead concentrations for individuals with bone loss were marginally higher than those for individuals with normal bone mineral density. No difference was found in plasma homocysteine concentrations between individuals with bone loss and individuals with normal bone mineral density even stratified by the combination groups of plasma folate and vitamin B_12_ (Table 3).

Subsequently, we analyzed the association between total urinary arsenic, blood lead, and plasma homocysteine concentrations and bone loss in the three groups. We found that in the low/low group an increase in blood lead concentrations was associated with an increase in the OR of bone loss (Table 4).

## 4. Discussion

In this study, we found aging to be associated with bone loss. Alcohol and coffee consumption increased and decreased the OR of bone loss, respectively. Total urinary arsenic and blood lead concentrations were positively correlated with the OR of bone loss in a dose–response manner. No correlation was observed between bone loss and plasma folate or vitamin B_12_ levels. However, this was the first observational study to find that an increase in blood lead concentrations was associated with an increase in the OR of bone loss in individuals with low plasma folate and plasma vitamin B_12_ levels.

In a previous study, arsenic was found in in vitro and in vivo experiments to directly and indirectly affect bone remodeling, respectively, which was mainly due to changes in osteoblast differentiation and function, resulting in a decrease in bone mineral density [20]. In another study, long-term exposure to low doses of lead reduced bone density and the number of cancellous bone trabeculae in male mice, which then inhibited bone formation and led to bone damage [21]. A cross-sectional study of the general population in the United States found that lead exposure was associated with a decrease in femoral and spine bone mineral density in premenopausal women [22]. In this study, we found a significant dose–response relationship between total urinary arsenic and blood lead concentrations and bone loss in adult and older patients who underwent health examinations. Arsenic and lead exposure were both related to bone loss.

Low blood lead concentrations have been found to interfere with the calcium regulation process; in an in vivo experiment involving lead-intoxicated animals, lead was observed to replace calcium and combine with osteocalcin, which resulted in a low bone formation rate [23]. Mineralization of bone tissue as a response to lead exposure may be mediated by changes in the bone turnover mechanism involved in mineral remodeling. These effects may be related to the adverse effects of lead on the processes regulating the bone turnover mechanism and affecting bone maturation and bone growth [24]. Bone can accumulate pentavalent arsenic for long periods; the accumulation of arsenate in bone may be due to the similarity between arsenate and phosphate. Arsenate may replace phosphate in hydroxyapatite crystals [25], which affects the function of bones.

A high serum total homocysteine concentration has also been reported to affect bones; this phenomenon occurs due to the deficiency of vitamin B_12_ and folate related to their metabolization, which is further related to decreased bone mineral density [26]. Previous studies have explored correlations among homocysteine, vitamin B_12_, folate, and bone mineral density; however, the results have been inconsistent. One study found that bone mineral density had a significantly positive correlation with serum folate levels, but not with homocysteine or vitamin B_12_, in postmenopausal women [27]. In addition, osteoporosis in Turkish postmenopausal women was reported to be related to homocysteine levels above the median and vitamin B_12_ values below the lowest quintile [28]. In healthy Moroccan postmenopausal women, high homocysteine and high vitamin B_12_ levels were reported to be independent risk factors for osteoporosis [29]. Furthermore, a meta-analysis indicated that high homocysteine and vitamin B_12_, but not folate, levels were associated with osteoporosis in postmenopausal women [11]. A 2021 study in American adults found that serum folate levels were positively correlated with bone mineral density; furthermore, although no correlation was found between serum vitamin B_12_ and bone mineral density, differences related to race and ethnicity were found [12]. In the present study, we observed that higher plasma folate and vitamin B_12_ levels were significantly related to lower plasma homocysteine levels. However, plasma folate, vitamin B_12_, and homocysteine levels were not related to bone loss. This may be due to the subjects in this study being patients undergoing health examinations rather than those with serious health problems. However, high blood lead concentrations were observed to be related to bone loss in the group with both low plasma folate and low vitamin B_12_ concentrations. A lack of vitamin B_12_ or taurine is often associated with delayed growth and bone maturation; in one study, a vitamin B_12_ deficiency in mice resulted in a significant reduction in body growth and a decline in bone mass due to the decreased activity of osteoblasts [30]. In addition, folate is a cofactor of nitric oxide synthase; it promotes the maintenance of bone density by contributing to the maintenance of optimal nitric oxide synthase activity in bone cells [31]. However, insufficient levels of serum folate prevent the appropriate regulation of fat and protein metabolization, which increases fatty acid synthesis and reduces muscle growth and function [32]. In addition, folate promotes cell homeostasis; therefore, an insufficient dietary intake of folate may prevent the counteraction of inflammation, apoptosis, and autophagy [32]. Moreover, low levels of plasma folate cannot indirectly regulate bone metabolism through bone–muscle crosstalk. The potential of low plasma folate and vitamin B_12_ levels caused by environmental metal exposure to increase the risk of bone loss requires further investigation.

Regarding the relationship between homocysteine levels and bone loss, a study of postmenopausal women found a significant negative correlation between homocysteine levels and bone loss [26]; another study reported that low plasma homocysteine levels were related to increased forearm bone mineral density [33]. However, other studies have found that homocysteine was not related to bone loss in postmenopausal women [34,35]. In this study, homocysteine levels in the low plasma folate and vitamin B_12_ group were significantly higher than those in the high plasma folate and vitamin B_12_ group; however, homocysteine levels were not related to bone loss. This may be because most of the participants in this study were healthy, and plasma homocysteine was, therefore, not involved in the process of bone loss. However, this topic requires further exploration.

This study has several limitations. Firstly, this was a cross-sectional study; therefore, the causal relationship between blood lead and total urinary arsenic concentrations and bone loss could not be confirmed. Biospecimens were collected only once for evaluation of blood cadmium and lead, total urinary arsenic, plasma folate, and plasma vitamin B_12_ concentrations. However, if all subjects have stable lifestyles and steady-state metabolism, these measurements may be reliable. Additionally, physical activity and dietary patterns could not be adjusted in the models, as these factors were not measured in our study. The study had the further limitations of a small sample size and the inability to obtain data on other factors that may affect bone mineral density. However, the results of this study revealed high blood lead levels to be associated with bone loss in participating individuals with low plasma folate and vitamin B_12_ levels.

## 5. Conclusions

Our study demonstrated that age increment and high blood lead and total urinary arsenic levels significantly increased the OR for bone loss in a dose–response manner. In addition, this study is the first observational study to find that an increase in blood lead concentrations tended to increase the OR of bone loss in a low plasma folate and low plasma vitamin B_12_ group. These findings indicate that for those with lower plasma folate and vitamin B_12_ concentrations, the greater the lead exposure, the higher the OR of bone loss.

## Figures and Tables

**Figure 1 nutrients-14-00911-f001:**
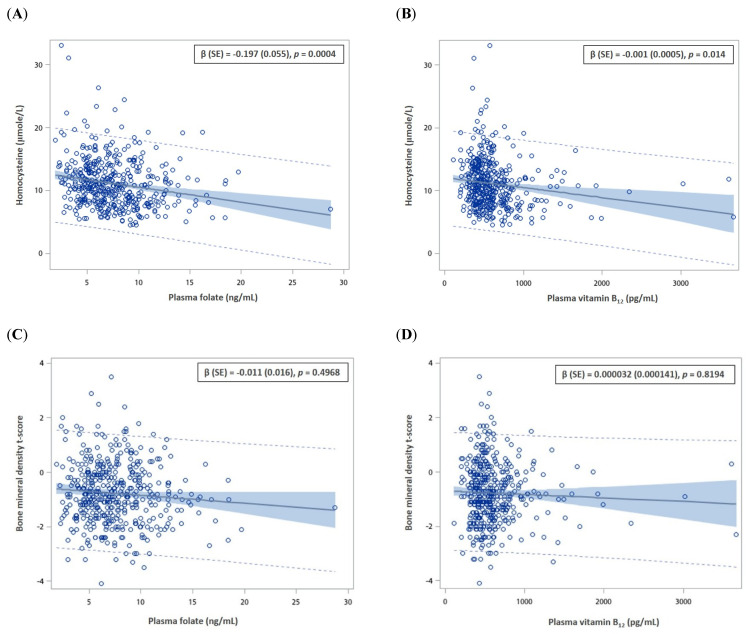
The correlations between plasma folate, vitamin B_12_, homocysteine, and bone mineral density while adjusting for age, sex, alcohol, and coffee consumption. (**A**) plasma folate and homocysteine; (**B**) plasma vitamin B_12_ and homocysteine; (**C**) plasma folate and bone mineral density; (**D**) plasma vitamin B_12_ and bone mineral density; (**E**) plasma vitamin B_12_ and folate.

**Table 1 nutrients-14-00911-t001:** The association between sociodemographic characteristics, lifestyle, and disease history and bone loss.

Variables	Bone Loss (*N* = 185) *N* (%)	Normal Bone Mineral Density (*N* = 262) *N* (%)	Age–Sex Adjusted OR (95% CI)
Age (years)	56.02 ± 9.66 ***	51.85 ± 10.23 ***	
≤50	38 (20.54)	109 (41.60)	1.00 ^a,§,^***
>50–65	122 (65.95)	136 (51.91)	2.43 (1.55–3.81) ***
>65	25 (13.51)	17 (6.49)	4.00 (1.94–8.24) ***
Sex			
Male	106 (57.30)	178 (67.94)	1.00 ^b^
Female	79 (42.70)	84 (32.06)	1.37 (0.92–2.05)
BMI (kg/m^2^)			
≤24	119 (64.32)	149 (56.87)	1.00
>24–27	37 (20.00)	65 (24.81)	0.62 (0.38–1.02) ^+^
>27	29 (15.68)	48 (18.32)	0.72 (0.42–1.24)
Educational level			
Illiterate/elementary	37 (20.00)	28 (10.69)	1.00
Junior/senior high	56 (30.27)	73 (27.86)	0.79 (0.42–1.49)
College and above	92 (49.73)	161 (61.45)	0.65 (0.36–1.18)
Cigarette smoking			
Non-smoker	135 (72.97)	182 (69.73)	1.00
Smoker	50 (27.03)	79 (30.27)	1.06 (0.66–1.69)
Alcohol consumption			
Never	109 (58.93)	165 (62.98)	1.00 ^§,^**
Frequent	31 (16.76)	52 (19.85)	2.14 (1.27–3.61) **
Occasional	45 (24.31)	45 (17.18)	1.36 (0.75–2.26)
Frequent or occasional	76 (41.07)	97 (37.03)	1.70 (1.10–2.63) *
Coffee consumption			
No	103 (55.68)	114 (43.51)	1.00
Frequent	45 (24.32)	94 (35.88)	0.81 (0.49–1.35)
Occasional	37 (20.00)	54 (20.61)	0.54 (0.34–0.85) **
Frequent or occasional	82 (44.32)	148 (56.49)	0.64 (0.43–0.94) *
Tea consumption			
No	76 (41.08)	82 (31.64)	1.00
Frequent	67 (36.22)	132 (51.56)	0.65 (0.42–1.01) ^+^
Occasional	42 (22.70)	43 (16.80)	1.29 (0.75–2.22)
Frequent or occasional	109 (58.92)	175 (68.36)	0.81 (0.54–1.21)
Diabetes			
No	173 (93.51)	243 (92.75)	1.00
Yes	12 (6.49)	19 (7.25)	0.67 (0.31–1.46)
Hypertension			
No	149 (89.54)	206 (79.54)	1.00
Yes	36 (19.46)	53 (20.46)	0.76 (0.47–1.26)

Abbreviations: OR, odds ratio; CI, confidence interval; BMI, body mass index. Values are expressed as the mean ± standard deviation. ^a^ Adjusted for sex. ^b^ Adjusted for age. ^§^
*p*-values < 0.05 for trend test. ^+^ 0.05 < *p* < 0.1, * *p* < 0.05, ** *p* < 0.01, *** *p* < 0.001.

**Table 2 nutrients-14-00911-t002:** The associations between total urinary arsenic, red blood cell lead and cadmium, plasma folate, vitamin B_12_, homocysteine, and bone loss.

Variables	Bone Loss *N* (%)	Normal Bone Mineral Density *N* (%)	Age–Sex Adjusted OR (95% CI)	Multivariate Adjusted OR (95% CI)
Total urinary arsenic (μg/g creatinine)	14.33 (0.13, 62.46) ^a,^**	11.25 (0.06, 92.99) ^a,^**		
≤9.30	41 (22.16)	87 (33.21)	1.00 ^§,^*	1.00 ^§,+^
>9.30–14.91	63 (34.05)	88 (33.58)	1.39 (0.84–2.31)	1.27 (0.76–2.12)
>14.91	81 (43.78)	87 (33.21)	1.72 (1.05–2.81) *	1.55 (0.94–2.56) ^+^
Blood lead (μg/dL)	4.91 (1.30, 19.25) ^a,^*	4.29 (0.92, 14.97) ^a,^*		
≤33.82	44 (23.78)	88 (33.58)	1.00 ^§,^**	1.00 ^§,^*
>33.82–56.14	66 (35.68)	87 (33.21)	1.60 (0.97–2.62) ^+^	1.52 (0.92–2.53) *
>56.14	75 (40.54)	87 (33.21)	1.95 (1.20–3.20) **	1.82 (1.10–3.01) *
Blood cadmium (μg/L)	1.26 (0.08, 10.86)	1.16 (0.12, 14.82)		
≤0.84	54 (29.19)	89 (33.97)	1.00	1.00
>0.84–1.58	64 (34.59)	87 (33.21)	1.06 (0.65–1.73)	1.00 (0.61–1.64)
>1.58	67 (36.22)	86 (32.82)	1.16 (0.72–1.89)	0.97 (0.59–1.61)
Plasma folate (ng/mL)	6.98 (2.28, 28.70)	6.95 (1.87, 18.50)		
≤5.83	55 (30.39)	88 (33.58)	1.00	1.00
>5.83–8.46	59 (32.60)	87 (33.21)	0.87 (0.54–1.41)	0.84 (0.52–1.36)
>8.46	67 (37.02)	87 (33.21)	0.70 (0.43–1.16)	0.71 (0.43–1.17)
Plasma vitamin B_12_ (pg/mL)	530.0 (113.0, 3666.0)	522.50 (198.0, 3608.0)		
≤442	58 (31.35)	87 (33.21)	1.00	1.00
>442–610	71 (38.38)	88 (33.58)	1.23 (0.77–1.96)	1.24 (0.77–1.99)
>610	56 (30.27)	87 (33.21)	0.85 (0.52–1.38)	0.87 (0.53–1.42)
Homocysteine (μmole/L)	10.52 (4.99, 31.06) ^a,+^	10.80 (4.50, 33.10) ^a,+^		
≤9.03	63 (34.05)	89 (33.79)	1.00	1.00
>9.03–12.78	80 (43.24)	87 (33.21)	1.31 (0.82–2.07)	1.34 (0.84–2.15)
>12.78	42 (22.70)	86 (32.82)	0.68 (0.41–1.15)	0.70 (0.41–1.19)

Abbreviations: OR, odds ratio; CI, confidence interval. Values are expressed as the median (minimum, maximum). The multivariate regression model was adjusted for age, sex, alcohol, and coffee consumption. ^a^ Wilcoxon rank-sum test. ^§^ Test for trend. ^+^ 0.05 < *p* < 0.1, * *p* < 0.05, ** *p* < 0.01.

**Table 3 nutrients-14-00911-t003:** Comparison of total urinary arsenic level, red blood cell lead, and plasma homocysteine between bone loss cases and normal bone mineral density stratified by a combination of plasma folate and vitamin B_12_ levels.

Variables	Overall	Bone Loss	Normal Bone Mineral Density
Low/low group for plasma folate and vitamin B_12_ (N = 141)			
Total urinary arsenic (μg/g creatinine)	12.97 (1.18, 62.46)	15.22 (4.77, 62.46) ^a,^*	11.10 (1.18, 52.68) ^a,^*
Blood lead (μg/dL)	4.79 (1.10, 14.51)	5.20 (1.31, 14.49) ^d,+^	4.50 (1.10, 14.51) ^d,+^
Plasma homocysteine (μmole/L)	11.42 (5.53, 31.06) ^e,^*	11.30 (5.53, 31.06)	11.40 (6.21, 26.34)
Low/high or high/low groups for plasma folate and vitamin B_12_ (N = 164)			
Total urinary arsenic (μg/g creatinine)	12.69 (0.13, 54.90)	14.57 (0.13, 44.0) ^b,+^	11.38 (3.35, 54.90) ^b,+^
Blood lead (μg/dL)	4.74 (0.92, 19.25)	5.08 (1.44, 19.25)	4.31 (0.92, 13.40)
Plasma homocysteine (μmole/L)	10.82 (4.99, 33.10) ^f,^*	10.49 (4.99, 18.81)	10.49 (5.62, 33.11)
High/high group for plasma folate and vitamin B_12_ (N = 142)			
Total urinary arsenic (μg/g creatinine)	12.26 (0.06, 92.99)	13.32 (2.53, 35.66) ^c,+^	11.51 (0.06, 92.99) ^c,+^
Blood lead (μg/dL)	4.46 (1.13, 16.23)	4.67 (1.30, 16.23)	4.24 (1.13, 14.97)
Plasma homocysteine (μmole/L)	9.44 (4.50, 24.40) ^e,f,^*	8.84 (5.19, 16.38) ^g,^*	9.87 (4.50, 24.40) ^g,^*

The low/low group was defined as participants with low levels of plasma folate (≤6.95 ng/mL) and vitamin B_12_ (≤522.50 pg/mL); the low/high and high/low groups were defined as participants either with a low level of plasma folate and a high level of vitamin B_12_ or a high level of plasma folate and a low level of vitamin B_12_; participants with high levels of plasma folate and vitamin B_12_ were defined as the high/high group. Values are expressed as the median (minimum, maximum). The comparison of the average of metals between three groups (overall) and between two groups (bone loss and normal mineral bone density) was determined by Kruskal–Wallis and Wilcoxon tests, respectively. The same English letters indicate that there was a significant difference between each of two groups. ^+^ 0.05 < *p* < 0.1, * *p* < 0.05.

**Table 4 nutrients-14-00911-t004:** Dose−response relationship between total urinary arsenic level, red blood cell lead, plasma homocysteine, and bone loss stratified by a combination of plasma folate and vitamin B_12_ levels.

Variables	Case/Control		Age–Sex Adjusted ORs (95% CI)	Multivariate Adjusted ORs (95% CI) ^a^
Low/low group for plasma folate and vitamin B_12_ (*N* = 141)				
Total urinary arsenic (μg/g creatinine)	Low	13/29	1.00 ^§,^^+^	1.00
	Moderate	15/28	1.33 (0.52–3.37)	1.17 (0.45–3.02)
	High	28/28	2.10 (0.89–4.94) ^+^	1.97 (0.82–4.97)
Blood lead (μg/dL)	Low	7/29	1.00	1.00
	Moderate	30/28	4.44 (1.65–11.97) **	4.24 (1.53–11.73) **
	High	19/28	2.64 (0.94–7.42) ^+^	2.44 (0.85–6.96) ^+^
Plasma homocysteine (μmole/L)	Low	19/29	1.00	1.00
	Moderate	29/29	1.38 (0.62–3.09)	1.31 (0.58–2.96)
	High	8/27	0.44 (0.16–1.21)	0.45 (0.16–1.26)
Low/high or high/low groups for plasma folate and vitamin B_12_ (*N* = 164)				
Total urinary arsenic (μg/g creatinine)	Low	16/30	1.00	1.00
	Moderate	27/33	1.34 (0.59–3.01)	1.32 (0.37–3.06)
	High	28/30	1.54 (0.68–3.47)	1.43 (0.62–3.29)
Blood lead (μg/dL)	Low	18/31	1.00 ^§,^^+^	1.00
	Moderate	23/32	1.33 (0.59–3.98)	1.19 (0.52–2.73)
	High	30/30	2.09 (0.93–4.66) ^+^	1.82 (0.80–4.15)
Plasma homocysteine (μmole/L)	Low	20/32	1.00	1.00
	Moderate	31/31	1.77 (0.81–3.85)	1.77 (0.81–3.92)
	High	20/30	1.17 (0.50–2.74)	1.21 (0.51–2.89)
High/high group for plasma folate and vitamin B_12_ (*N* = 142)				
Total urinary arsenic (μg/g creatinine)	Low	9/27	1.00	1.00
	Moderate	27/29	2.52 (0.96–6.63) ^+^	2.63 (0.96–7.24) ^+^
	High	22/28	1.85 (0.68–5.03)	1.61 (0.57–4.57)
Blood lead (μg/dL)	Low	15/28	1.00	1.00
	Moderate	24/29	1.35 (0.57–3.20)	1.48 (0.59–3.67)
	High	19/27	1.73 (0.69–4.36)	1.73 (0.65–4.56)
Plasma homocysteine (μmole/L)	Low	27/29	1.00	1.00
	Moderate	19/28	0.65 (0.28–1.49)	0.72 (0.30–1.71)
	High	12/27	0.41 (0.16–1.06) ^+^	0.42 (0.16–1.10) ^+^

The low/low group was defined as participants with low levels of plasma folate (≤6.95 ng/mL) and vitamin B_12_ (≤522.50 pg/mL); the low/high and high/low groups were defined as participants either with a low level of plasma folate and a high level of vitamin B_12_ or a high level of plasma folate and a low level of vitamin B_12_; participants with high levels of plasma folate and vitamin B_12_ were defined as the high/high group. Cut-off points for total urinary arsenic (μg/g creatinine), blood lead (μg/dL) and plasma homocysteine (μmole/L) were 9.38 and 14.67 (μg/g creatinine), 28.98 and 59.46 (μg/dL), and 10.19 and 13.75 (μmole/L), respectively, in the low/low group; 0.09456 and 0.15835 (μg/g creatinine), 3.572 and 5.518 (μg/dL), and 8.9 and 12.8 (μmole/L), respectively, in the low/high and high/low groups; 0.08439 and 0.14905 (μg/g creatinine), 3.346 and 5.982 (μg/dL), and 8.44 and 11.73 (μmole/L), respectively, in the high/high group. ^+^ 0.05 ≤ *p* < 0.1. ** *p* < 0.01. ^§^
*p*-values < 0.05 for trend test. ^a^ Adjusted for age, sex, alcohol, and coffee consumption.

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
