# Peer review of "Associations between Plasma Folate and Vitamin B12, Blood Lead, and Bone Mineral Density among Adults and Elderly Who Received a Health Examination"

_nutrients, 2022, doi:10.3390/nu14040911_

Round 1

Reviewer 1 Report

In this study the authors investigated the whether vitamin B12 and folate modify the association between environmental metal exposure (lead, cadmium and arsenic) and bone loss in the adult population (over 50) in Taiwan.

The aim of this study is interesting, however, I have the following comments and questions of the manuscript:

Whether was the sample size estimated?

More information should also be provided on selection of participants e.g. inclusion/exclusion criteria  for the study.

Important details on participants are lacking, e.g. age range, data on the use of dietary supplements?

Was the physical activity of the respondents assessed? - it is important in the context of BMD.

The conditions for blood collection for tests should also be added in the methods section: time of day, hours, were they the same for all the subjects?

Were the blood tests done immediately or whether the blood (plasma) was frozen in the same way as urine - if so, add it in manuscript.

In the case of BMD measurement no specifications were given for the equipment used.

All data were presented as mean values and SD. Did all data have normal distribution?

The results should be carefully read again, some data are repeated in the text - are they needed?

Table 4 is hardly readable - it can be placed horizontally - for consideration.

Whether all data were normally distributed, only means and SD are shown?

The first sentence in the discussion concerns the influence of age - this is not included in the conclusions later - it should be harmonized.

I propose to add in the limitations these other factors not examined - list them.

In general, the work is very interesting and the topic is worth further research, which I strongly encourage the Authors to do.

Reviewer 2 Report

This article presents an observational study showing that blood lead concentrations tend to increase the OR of bone loss in the group with low plasma folate and vitamin B12. The study is well written, results are clearly and adequately analyzed and the conclusion is supported by the data presented. English language should be corrected lines 69 and 74.
